# Economic Burden of Cervical and Head and Neck Cancer in Taiwan from a Societal Perspective

**DOI:** 10.3390/ijerph20043717

**Published:** 2023-02-20

**Authors:** Ying-Hui Wu, Chyong-Huey Lai, Ling Chien, Yun-Chung Pan, Yu-Jr Lin, Casey Feng, Chee-Jen Chang

**Affiliations:** 1MSD Taiwan, Taipei 11047, Taiwan; 2Gynecologic Cancer Research Center, Chang Gung Memorial Hospital, Linkou Branch, Taoyuan 33305, Taiwan; 3Department of Obstetrics and Gynecology, Chang Gung Memorial Hospital, Linkou Branch and Chang Gung University, Taoyuan 33305, Taiwan; 4Research Service Center for Health Information, Chang Gung University, Taoyuan 33302, Taiwan; 5Clinical Informatics and Medical Statistics Research Center, Chang Gung University, Taoyuan 33302, Taiwan; 6Graduate Institute of Clinical Medical Science, Department of Biomedical Science, Chang Gung University, Taoyuan 33302, Taiwan

**Keywords:** human papillomavirus, head and neck cancer, cervical cancer, socioeconomic burden

## Abstract

Background: Head and neck cancers (HNC) are increasingly recognized as important human papillomavirus (HPV)-related malignancies in addition to cervical cancer (CC). However, data on the socioeconomic impact of HNC and CC in Taiwan are limited. Methods: A retrospective cohort study was conducted to estimate the total direct medical cost and indirect productivity loss from CC and HNC between 2014 and 2015. Patient data from the Taiwan National Cancer Registry were analyzed, with matched non-cancer controls from the Taiwan National Healthcare Reimbursement Database. Indirect costs due to premature deaths were calculated using public data from Taiwanese government reports. Results: In the direct cost analysis, 2083 patients with newly diagnosed CC and 11,078 with newly diagnosed HNC (10,036 males) were identified between 2014 and 2015 and followed up through the end of 2016 or until death. The total direct medical costs incurred in 2014 and 2015 due to HNC were 11.54 times higher in males than in females, and 4.55 times higher than CC. Indirect cost analysis showed the total annual productivity loss was New Taiwan Dollar (NTD) $12 billion in 2019, and 79.99% was attributed to male HNC. Conclusion: In Taiwan, the socioeconomic burden associated with male HNC is high and greater than that seen with CC. While not all HNCs are attributable to HPV infection, prevention of HNC through HPV vaccination should be considered for both sexes.

## 1. Introduction

Human papillomavirus (HPV)-related diseases include benign conditions such as recurrent respiratory papillomatosis and anogenital and cutaneous warts, and malignant diseases, including cervical cancer (CC), male and female genital cancers, and head and neck cancers (HNC) [1].

Cervical cancer is the most well-known HPV-related cancer, the fourth most common female malignant tumor worldwide, and the ninth most diagnosed cancer among females in Taiwan [2,3]. The 2019 Taiwan Cancer Registry Annual Report revealed an age-standardized incidence rate of 29.07 cases per 100,000 person-years, composed of 21.41 cases per 100,000 person-years for carcinoma in situ and 7.85 cases per 100,000 person-years for invasive CC [2]. The age-standardized mortality rate for CC was 3.20 per 100,000 person-years [2].

Two interventional measures have been implemented to control CC in Taiwan. Firstly, since 1995, an annual Pap cervical screening service has been offered to females aged 30 and above. The screening program has greatly reduced the absolute mortality of CC by approximately 50% from the early 1990s to the late 2000s [4]. Secondly, in 2018, the HPV vaccination program was initiated for Taiwanese school-age females to prevent CC. Evaluating the outcome of this program will require more time.

In addition to CC, HPV is a causative pathogen for many other types of cancer, among which HNCs (including oral, oropharyngeal, and hypopharyngeal cancers) are the most common. In the United States, the number of patients with oropharyngeal cancer attributable to HPV has exceeded those with CC [5]. In Denmark, the absolute number of newly diagnosed HPV-positive patients with oropharyngeal squamous cell carcinoma is expected to surpass that of CC in 2016 [6]. Compared with other regions in the world, Taiwan has higher incidence (22.57 vs. 7.2 per 100,000 person-years) and mortality rates (8.86 vs. 3.42 per 100,000 person-years) for HNC [2,7]. In addition, 37.05% of patients with oral cancer were diagnosed with stage IV disease, with a poor prognosis [2]. Other than poor clinical outcomes, HNCs are also associated with a huge psychosocial impact. For example, patients with HNC have a 3–fold higher incidence of depressive disorders and a 1.9-fold higher risk of suicide attempts compared with patients with other types of cancer [8,9]. Nevertheless, preventive measures for HPV-related HNC are limited.

Existing studies showing the economic burden of CC and HNC are limited to direct medical costs, which underestimate the burden of both cancers on society since they affect females of childbearing age, and both females and males during the age of active employment in the workforce. Furthermore, there has been limited research on the direct or indirect costs of either CC or HNC in Taiwan. Hence, this study aimed to estimate the economic burden, accounting for both direct and indirect costs, of CC in females and HNC in males and females in Taiwan.

## 2. Materials and Methods

This retrospective cohort study was conducted according to the guidelines of the Declaration of Helsinki, and approved by the Institutional Review Board of Chang Gung Memorial Hospital (protocol code 202000296B0 [2002170044]; 12 March 2020). Informed consent from the participating patients was waived by Institutional Review Board of Chang Gung Memorial Hospital as this was a secondary database study. All cost estimates are represented in NTD. The average exchange rate in 2019 was 1 NTD to 0.0324 USD.

### 2.1. Estimation of Direct Medical Costs

Direct medical costs included all expenses from healthcare services, medication, and diagnostic and therapeutic interventions incurred during medical visits and hospitalization. In the estimation of direct medical costs, patients with newly diagnosed CC or HNC were identified from the Taiwan National Cancer Registry during 2014–2015 and linked to the Taiwan National Healthcare Reimbursement Database (NHRD) for cost estimation with an identity number shared in both databases. All hospitals with more than 50 beds are required to report to the National Cancer Registry upon diagnosis. Thus, it covers 98% of patients with cancer in Taiwan [10]. Matched non-cancer control patients were selected from the NHRD during the same time period by propensity score matching. The propensity score consists of sex, age, Charlson comorbidity index, and the top ten diseases with highest medical expenses in the NHRD [11]. Newly diagnosed cancer was defined as the existence of cancer-related ICD-0-3 codes during the study period, and comorbidities were recognized by the existence of ICD9/ICD10 codes (Appendix A) in at least one hospitalization or two clinic visits. Patients with unknown cancer stages in the registry or with a history of malignancy were excluded from the analysis.

Since most of the medical expenses for patients with cancer typically occur in the initial treatment period and during end-of-life management prior to death, we report direct medical costs in the following three periods: 120 days after diagnosis of cancer (post-diagnosis), 90 days before death (pre-mortality), and in-between. All costs were standardized per person-year to enable comparison between cancers and non-cancer controls. For patients with cancer who survived less than 90 days after diagnosis, all the costs were categorized into the pre-mortality period, whereas patients who survived 100 days after diagnosis, the last 90 days were counted in pre-mortality expense and the rest categorized into post-diagnosis period. The 2019 consumer price index was applied to align with the indirect cost estimation.

### 2.2. Estimation of Indirect Costs

Indirect costs were defined as productivity loss due to cancer-related premature death. For estimation of life expectancy, human capital approach was used to estimate the lost productivity that would otherwise accrued in 2019 for men and women dying from CC and HNC [12,13]. We reviewed the number of deaths due to CC from 1970, and due to HNC from 1974 using Vital Statistical Abstract (1970–1971), Health Statistics—Vital Statistics (1972–1979), and statistical data reported by the Ministry of Health and Welfare in Taiwan (1980–2019) [14]. Of note, the number of deaths from HNC in 1972 and 1973 was missing in the governmental statistical report, so the analysis started in 1974 to maintain data continuity. In addition, historical life tables from 1970 to 2019 were reviewed to acquire the life expectancy in sex and age-specific groups in each year. The predicted age of patients who died from CC or HNC, but would have been expected to live until 2019 according to life tables, was calculated. For example, if a 40-year-old female died from CC in 2015, and the life expectancy for females aged 40 in that year was 78 years, then the patient was included in the analysis with an age of 44 years in 2019. Since only data up to 2019 were available, we derived the annual indirect cost of CC and HNC in 2019 by multiplying the loss of human capital per person and life expectancy in 2019 if these patients had not died from CC or HNC. The ICD-9/ICD-10 codes for CC and HNC included in the indirect cost analysis are summarized in Appendix A.

For estimation of human capital loss, labor force participation rate, unemployment rate, and median annual earnings were retrieved from the statistical data of the Taiwan Ministry of Labor, with stratification to 5-year age groups (Appendix A) [15,16,17]. To derive indirect costs during 2019, we multiplied the estimated number of people who would otherwise have been alive within each 5-year age group with human capital loss in each age group.

### 2.3. Statistical Analysis

Two-tailed independent sample t-tests for the continuous variables and Chi-square test for the categorical variables were implemented to compare the differences between patients with cancer and matched controls without cancer, with a significance level of 0.05. All statistical analyses were conducted using SAS (version 9.4) or R (version 3.5.2).

## 3. Results

### 3.1. Direct Cost

In 2014–2015, 2083 patients with newly diagnosed CC, and 11,078 patients with newly diagnosed HNC were identified. After excluding patients with unknown staging, 1996 patients with CC and 10,278 patients with HNC were matched to the same number of non-cancer controls (1996 for CC and 10,278 for HNC, respectively). Among the patients with HNC, 9306 (91%) were male and 972 (9%) were female. The demographics and comorbidities of patients with cancer and non-cancer controls are listed in Table 1. The prevalence of comorbidities was not statistically significantly different, except that the proportion of patients with comorbid hypertension was lower in the CC group than in controls (29% vs. 33%; *p* = 0.009).

The aggregated direct medical cost of CC and HNC per year is summarized in Table 2. HNC in males resulted in direct medical costs that were 11.54 times higher than those seen with HNC in females. The direct medical cost of HNC in males was 4.55 times the direct medical cost of CC in females. In addition, as the cancer stage advanced, increased frequencies of visiting outpatient clinics and ER, along with frequency and length of hospitalization, were observed.

Figure 1 illustrates the per person-year direct medical costs in CC, male HNC, and female HNC for each designated period (120 days after diagnosis of cancer, 90 days before death, and in-between), and in matched non-cancer controls. Compared with non-cancer controls, patients with CC had direct per person-year costs 42-, 48-, and 4-times higher at 120 days after diagnosis of cancer, 90 days before death, and in-between, respectively. Similarly, female HNC resulted in 37-, 33-, and 5-fold increases in per person-year direct medical costs compared with non-cancer controls, whereas male HNC resulted in 59-, 43-, and 8-fold increases in the designated periods. Among the three cancer groups, male HNC incurred the highest per person-year direct medical costs.

### 3.2. Indirect Costs

Figure 2 demonstrates the number of people who would have been alive in 2019 had they not died from HNC or CC between the 1970s and 2019. The number of males who died from HNC and would have been alive in 2019 was notably higher than the number of females who died from HNC and CC and would have been alive in 2019. The age distribution mainly fell between 60 and 70 years old for both sexes (male, 36.37%; female, 32.7%).

Annual productivity loss due to male HNC, female HNC, and CC-related mortality in 2019 is summarized in Table 3. In total, the annual productivity loss (i.e., the annual indirect cost) of these cancers was NTD $12,026,819,000, of which 79.99% was attributed to male HNC, greater than female HNC (8.14%) and CC (11.86%). The cost was primarily attributable to the loss of population that would have been 50–65 years old if alive in 2019.

## 4. Discussion

This was the first real-world study to estimate the direct and indirect medical costs of CC and HNC using national claim and cancer registry data in Taiwan. The total direct medical costs for CC and HNC throughout 2014 and 2015 was NTD $9.9 billion, which represents a large financial burden for Taiwanese society and is equivalent to almost 26 times the NTD $380 million vaccine budget for the 2022 HPV national immunization program [17].

Aside from the direct medical costs, our study also highlighted productivity loss due to death in working age adults (indirect cost) caused by CC and HNC. This approach differed from previously pharmaco-economic studies that usually focus on direct medical costs alone. In our study, it was estimated that 60,354 males and 20,945 females would be alive in 2019 had they not died from HNC or CC between the 1970s and 2019, leading to an annual indirect productivity loss of NTD $12,026,819,000 in 2019, an estimation that may better reflect actual societal impact. Even though the ICD codes used in the indirect cost analysis were different from the ICD code used in the direct cost analysis, according to our estimation, the resulting difference in the number of patients who died in each analysis is small (within 8%) compared with the total number. Therefore, our data on economic loss due to premature death from potentially preventable diseases are likely to be meaningful.

This study divided the direct medical expenses associated with CC and HNC into three periods, post-diagnosis (120 days after diagnosis of cancer), pre-mortality (90 days before death), and in-between, with the majority of treatment costs occurring in the post-diagnosis and pre-mortality periods. The rationale for this approach was that the primary treatment of CC and HNC includes surgical resection followed by adjuvant radiotherapy or systemic therapy, all of which occur within around 90–120 days of diagnosis [18,19]. In addition, Taiwanese National Health Insurance requires that radiotherapy for inoperable CC should be completed in 9 weeks, and adjuvant concurrent chemoradiation for HNC patients should start from 4–6 weeks after surgery. The temporal division was also supported by a pharmaco-economic study on the costs of healthcare for patients with HPV-related cancer, which followed patients for 2 years after diagnosis, and concluded that most cancer costs were incurred during the first 6 months, and then stabilized [20]. The costs associated with the three periods were further standardized per person-year, which allowed comparison between different periods and groups (i.e., male HNC, female HNC, CC, and control). Our final data confirmed that the majority of medical costs were incurred within 120 days of cancer diagnosis and 90 days before death, justifying the selection of newly onset cancer cases.

Data on HPV attribution rates in cancer varies by anatomical site and geographic region [21]. In the United States, HPV accounts for more than 68% of oropharyngeal cancers and the number of patients with HPV-positive oropharyngeal squamous cell carcinoma has surpassed the number reported for CC [5,22]. Globally, the HPV attribution rate is about 25% in oropharyngeal cancer, 21.4% in pharyngeal cancer, and 7.4% in oral cavity cancer [21]. In Taiwan, data on HPV attribution rates in different HNC anatomic sites is limited, ranging from 12.6% in tonsil cancer to 24.9% in oropharyngeal cancer [23,24]. In contrast, the prevalence of HPV DNA in invasive carcinoma of the cervix was 100% [25]. While more research is needed to discover the true HPV attribution rate, the higher socioeconomic burden seen with HNC than with CC should not be overlooked. Compared with CC, HNC had a higher incidence, a later stage upon diagnosis with an associated increase in mortality rates, higher direct medical costs, and a greater indirect productivity loss. Overall, these data highlight the high socioeconomic burden of HNC compared with CC.

Cancer prevention is a major goal in public health globally. HPV vaccination has been demonstrated to be effective in preventing HPV persistent infection and subsequent cancer development [26,27]. Though the HPV program was introduced with a focus on CC prevention, in recent years, an increasing number of countries have adopted a gender-neutral approach. As of April 2019, 32 countries had recommended gender-neutral vaccination programs, and the number is growing. Most are western countries, including the United States, Germany, and Italy [28]. Our data show that the economic burden associated with HNC in males is 4.5 times higher in direct medical costs and 6.7 times higher in productivity loss compared with CC in females in Taiwan. It is warranted therefore that HPV-related HNC prevention should be a key consideration for long-term cancer prevention.

There are some limitations in this study. First, the observation time was short due to the limited accessibility of the data. Second, medical expenses may have been underestimated from registry data, since the increasingly utilized out-of-pocket modalities of diagnosis and treatment in the current rapid-evolving oncology area are often not reimbursed. For example, immunotherapy was not reimbursed in Taiwan before 2019. Third, as productivity loss in people who would otherwise be alive in 2019 was estimated from historical data, cause of death misclassifications may have occurred, leading to an underestimation of deaths from HPV-related cancers. Finally, HPV positivity data were not captured in either database or available publicly, which may be a concern, especially in betel nut endemic areas such as Taiwan. Chen et al. reported a HPV positivity rate of 25% among patients with oropharyngeal squamous cell carcinoma in a Taiwanese single-center retrospective study [24]. Compared with HPV-negative patients, the HPV-positive cases tend to be younger and have a higher socioeconomic status, suggesting a higher impact of productivity loss.

## 5. Conclusions

In conclusion, this study demonstrated a substantial socioeconomic burden associated with CC and HNC in Taiwan, in both direct medical costs and indirect productivity loss. Male HNC accounted for a greater economic burden than female HNC and CC, emphasizing the potential need for a gender-neutral HPV vaccination program.

## Figures and Tables

**Figure 1 ijerph-20-03717-f001:**
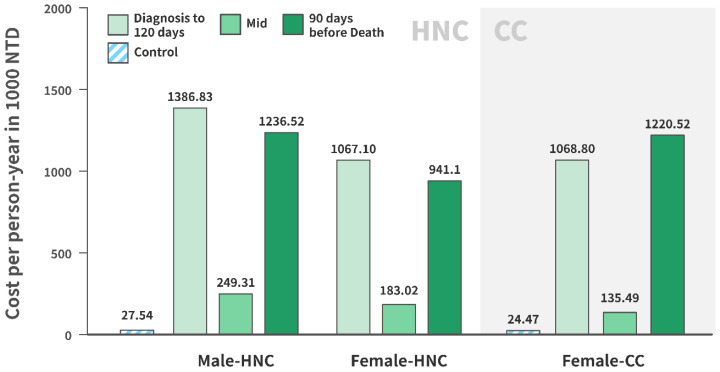
Per person-year direct medical costs of CC and HNC in 2014–2015 compared with non-cancer controls.

**Figure 2 ijerph-20-03717-f002:**
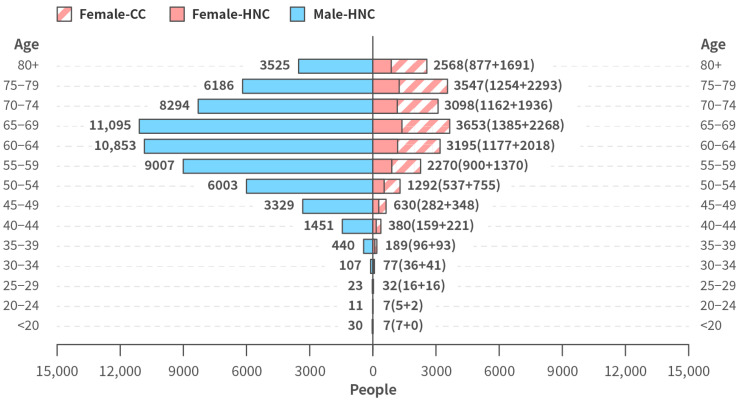
Age and sex distribution of patients who would otherwise be alive in 2019 had they not died from HNC or CC between 1970 and 2019.

**Table 1 ijerph-20-03717-t001:** Demographics and comorbidities of patients with newly diagnosed CC and HNC in 2014–2015 and in matched non-cancer controls.

		CC							HNC						
		Total		Control (N = 1966)		Patient (N = 1966)		*p*-Value	Total		Control (N = 10,278)		Patient (N = 10,278)		*p*-Value
		Mean	SD	Mean	SD	Mean	SD		Mean	SD	Mean	SD	Mean	SD	
Age		57.69	15.23	58.20	15.46	57.18	14.99	0.035	55.55	11.58	55.42	11.54	55.68	11.61	0.104
Frequency of visiting a dental clinic		1.00	2.02	0.98	1.95	1.02	2.08	0.564	1.30	2.48	1.30	2.40	1.30	2.56	0.955
		Count	%	Count	%	Count	%		Count	%	Count	%	Count	%	
Sex	Male	0	0	0	0	0	0		18596	90%	9290	90%	9306	91%	0.722
	Female	3932	100	1966	100	1966	100		1960	10%	988	10%	972	9%	
Acute myocardial infarction		19	0%	12	1%	7	0%	0.358	123	1%	58	1%	65	1%	0.587
Heart failure		79	2%	41	2%	38	2%	0.820	327	2%	154	1%	173	2%	0.316
Peripheral vascular disease		20	1%	10	1%	10	1%	1.000	171	1%	80	1%	91	1%	0.443
Cerebrovascular disease		220	6%	122	6%	98	5%	0.110	1188	6%	582	6%	606	6%	0.492
Dementia		81	2%	44	2%	37	2%	0.501	145	1%	62	1%	83	1%	0.096
Chronic lower respiratory disease		263	7%	129	7%	134	7%	0.798	1963	10%	982	10%	981	10%	1.000
Connective tissue disease		55	1%	21	1%	34	2%	0.103	110	1%	53	1%	57	1%	0.774
Peptic ulcer disease		375	10%	181	9%	194	10%	0.515	1954	10%	962	9%	992	10%	0.490
Chronic liver diseases		144	4%	77	4%	67	3%	0.445	1655	8%	824	8%	831	8%	0.878
Diabetes mellitus		503	13%	268	14%	235	12%	0.127	3519	17%	1769	17%	1750	17%	0.739
Diabetes mellitus with complications		111	3%	53	3%	58	3%	0.700	937	5%	475	5%	462	4%	0.688
Hemiplegia		13	0%	6	0%	7	0%	1.000	72	0%	35	0%	37	0%	0.906
Renal disease		177	5%	85	4%	92	5%	0.644	817	4%	416	4%	401	4%	0.617
Malignancy		54	1%	27	1%	27	1%	1.000	498	2%	249	2%	249	2%	1.000
Moderate to severe liver diseases		5	0%	<3	0%	<3	0%	1.000	34	0%	17	0%	17	0%	1.000
Metastatic cancer		<3	0%	<3	0%	<3	0%	1.000	<3	0%	<3	0%	<3	0%	1.000
HIV		0	0%	0	0%	0	0%	1.000	16	0%	7	0%	9	0%	0.803
Psychiatric disorder		33	1%	18	1%	15	1%	0.727	116	1%	55	1%	61	1%	0.642
Hypertension		1219	31%	648	33%	571	29%	0.009	6331	31%	3113	30%	3218	31%	0.116
Respiratory failure		42	1%	23	1%	19	1%	0.642	181	1%	89	1%	92	1%	0.081
Ischemic heart disease		253	6%	127	6%	126	6%	1.000	1612	8%	821	8%	791	8%	0.452

CC, cervical cancer; HIV, human immunodeficiency virus; HNC, head and neck cancer; SD, standard deviation.

**Table 2 ijerph-20-03717-t002:** Aggregated direct medical cost of newly diagnosed cancers in the National Health Insurance Research Database in a single year (NTD).

	2014			2015		
	CC	HNC	Total	CC	HNC	Total
Females	$893,201,718	$352,298,324	$1,245,500,042	$743,760,951	$281,322,088	$1,025,083,038
Males	0	$4,068,256,435	$4,068,256,435	0	$3,561,574,322	$3,561,574,322
Total	$893,201,718	$4,420,554,759	$5,313,756,477	$743,760,951	$3,842,896,410	$4,586,657,360

**Table 3 ijerph-20-03717-t003:** Estimated productivity loss in Taiwan due to CC and HNC mortality in 2019.

	Male		Female			
	HNC		HNC		CC	
Age Group	Number of Working People	Productivity Loss (NTD)	Number of Working People	Productivity Loss (NTD)	Number of Working People	Productivity Loss (NTD)
<20	0	$0 (0%)	1	$349,000 (0%)	0	$0 (0%)
20–24	6	$2,094,000 (0%)	3	$1,047,000 (0%)	1	$349,000 (0%)
25–29	20	$9,460,000 (0%)	14	$6,622,000 (1%)	14	$6,622,000 (0%)
30–34	96	$51,456,000 (1%)	32	$17,152,000 (2%)	37	$19,832,000(1%)
35–39	380	$203,680,000 (2%)	83	$44,488,000 (5%)	80	$42,880,000 (3%)
40–44	1178	$673,816,000 (7%)	129	$73,788,000 (8%)	179	$102,388,000 (7%)
45–49	2764	$1,581,008,000 (16%)	234	$133,848,000 (14%)	289	$165,308,000 (12%)
50–54	4378	$2,346,608,000 (24%)	392	$210,112,000 (21%)	545	$295,336,000 (21%)
55–59	4968	$2,662,848,000 (28%)	496	$265,856,000 (27%)	736	$405,216,000 (28%)
60–64	3898	$2,089,328,000 (22%)	423	$226,728,000 (23%)	692	$388,600,000 (27%)
Sum	17,688	$9,620,298,000	1807	$979,990,000	2573	$1,426,531,000

Mean annual income of an employed individual in Taiwan was NTD $650,000 in 2020.

## Data Availability

Statistics of Death: 1. Publicly available data of mortality statistics in cervical cancer (CC) and head and neck cancer (HNC) from 1980 to 2019 were analyzed in this study. This data can be found here: https://www.mohw.gov.tw/dl-41902-68a91b00-416c-4d0d-a02f-68a0725d9dcf.html, accessed on 28 November 2022. 2. Publicly available data of mortality statistics in cervical cancer (CC) and head and neck cancer (HNC) from 1970 to 1979 were analyzed in this study. This data can be found here: https://dep.mohw.gov.tw/DOS/lp-5066-113-xCat-003-2-20.html, accessed on 28 November 2022 (* Because the statistics of oropharyngeal cancer deaths in 1973–1974 were absent, the data of oropharyngeal cancer was retrieved from 1975 to maintain data continuity). Life expectancy: Publicly available data of life expectancy were analyzed in this study. This data can be found here: https://www.moi.gov.tw/cl.aspx?n=2981, accessed on 28 November 2022. Labor force Participation: Publicly available data of labor force participation in 2019 were analyzed in this study. This data can be found here: https://statfy.mol.gov.tw/index01.aspx, accessed on 28 November 2022. Unemployment Rate: Publicly available data of unemployment rate in 2019 were analyzed in this study. This data can be found here: https://statdb.mol.gov.tw/statis/jspProxy.aspx?sys=210&kind=21&type=1&funid=q02074&rdm=Ayjbaatf, accessed on 28 November 2022. Annual total salary: Publicly available data of annual total salary were analyzed in this study. This data can be found here: https://earnings.dgbas.gov.tw/template.html?target=1, accessed on 28 November 2022. Probability of death: Publicly available data of annual total salary were analyzed in this study. This data can be found here: https://www.moi.gov.tw/stat/node.aspx?cate_sn=&belong_sn=6189&sn=6190, accessed on 28 November 2022.

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
