# Peer review of "Economic Burden of Cervical and Head and Neck Cancer in Taiwan from a Societal Perspective"

_ijerph, 2023, doi:10.3390/ijerph20043717_

Round 1
Reviewer 1 Report
The paper tackled an important topic which could be of interest to health policy makers. The direct costs of the two cancers followed the usual methods of measuring direct costs of diseases which is good. The authors made good attempt in measuring the indirect costs. However, the approach to estimating the indirect costs needs further attention. Productivity loss was mentioned but with little or no assumptions or supporting theories for justification. This may contribute to errors in the calculation. For example, people who are 24 years or less in the sample had zero productivity loss reported. The use of 2019, 5-year age group bands or cohorts and the labour market information in the calculations need further explanation. It will be good to state the official retirement age, expected average years of education and income for the age cohorts and explain how such information were used or factored into the indirect cost calculations. Reporting the costs in local currency is good but the authors may improve interest and citation by worldwide audiences by including information about the convertibility of the local currency into one of the internationally dominant currencies using a dated exchange rate. The paper will benefit from professional proofreading.
Author Response
Point 1: However, the approach to estimating the indirect costs needs further attention. Productivity loss was mentioned but with little or no assumptions or supporting theories for justification. This may contribute to errors in the calculation. For example, people who are 24 years or less in the sample had zero productivity loss reported.
Thanks for your suggestions. The indirect cost analysis used human capital approach to estimate the lost productivity that would otherwise accrued in 2019 for men and women dying from CC and HNC, which have been mentioned or adopted in the following 2 literatures,
- Brown ML, Lipscomb J, Snyder C. The burden of illness of cancer: economic cost and quality of life. Annu Rev Public Health. 2001;22:91-113.
- Insinga RP. Annual productivity costs due to cervical cancer mortality in the United States. Womens Health Issues. 2006 Sep-Oct;16(5):236-42.
The information is also added in “Method and Material”
The productivity loss of people who would otherwise be alive and were 24-year-old or less in 2019 was included in analytic framework. However, the number of people was zero resulting in zero productivity loss.
Point 2: The use of 2019, 5-year age group bands or cohorts and the labour market information in the calculations need further explanation.
Thanks for your suggestions. The death statistic report by Taiwanese government was stratified by a 5-year age group by default.
Point 3: It will be good to state the official retirement age, expected average years of education and income for the age cohorts and explain how such information were used or factored into the indirect cost calculations.
The official retirement age is 65 years old. The labor participation rate, unemploument rate and median annual income in Taiwan in 2019 are summarized in Table S3.
Point 4: Reporting the costs in local currency is good but the authors may improve interest and citation by worldwide audiences by including information about the convertibility of the local currency into one of the internationally dominant currencies using a dated exchange rate.
It is added that “The average exchange rate in 2019 was 1NTD to 0.0324USD.” We also provide the average annual employment income as a reference point to consider.
Point 5: The paper will benefit from professional proofreading.
Professional English proofreading was done as suggested. Thank you.
Reviewer 2 Report
Dear Authors,
I read your article on the economic burden of cervical and HNC in Taiwan from a societal perspective with great interest.
Please find attached my detailed comments in the attached document.
My most important suggestions would be to add more information to Table 2 to also show direct costs by cancer stage and/or age groups and to clearly state what is included in 'direct medical costs' (e.g. patient out-of-pocket costs or only costs to health services).
I trust these changes will improve the clarity and quality of your manuscript further.
Kind regards,
Reviewer 1

Author Response
Thank you for your kind suggestions. We have revised the manuscript accordingly.
Point 1: Material & Methods
Please clarify how you avoided double-counting these costs and how these costs were counted in both periods
Thank you for the suggestions. Clarifications were made directly in the manuscript.
Point 2: Results
Could you add in cancer stages to Table 2 given these interesting differences observed which will be reflected in direct costs; or it could be listed by age groups in line with the illustration of indirect costs (Table 3)
Thank you for the suggestions. However, our access to the Taiwan National Healthcare Reimbursement Database expired. Therefore, additional analyses could not be provided. Thank you for understanding.
Point 3: Discussion
L195:Please explain why different ICD codes were used
Different ICD codes were used because the source databases were different. The ICD code in direct and indirect cost analysis are summarized in table S1 and table S2.